# Test Time Augmentations are Worth One Million Images for Out-of-Distribution Detection

## Abstract

Out-of-distribution (OOD) detection is commonly improved either by storing large in-distribution (InD) reference sets (e.g., nearest-neighbor methods) or by exposing the model to auxiliary OOD data during training. Both requirements limit deployability at scale. This paper shows that carefully chosen test-time augmentations (TTA) can provide a strong, self-referential signal for OOD detection from a *single* test input, without any stored InD data and without OOD exposure. We first identify a practical taxonomy that separates mild, feature-preserving *InD augmentations* (IDAs) from aggressive *OOD augmentations* (OODAs), and empirically demonstrate that IDAs consistently improve detection while OODAs often degrade it. Building on this insight, we propose a simple plug-and-play detector based on sequential masking: for each test image, we generate a small set of masked views and use the $k$-th largest embedding similarity to the original image as an "ID-ness" score. With only 25 TTAs per input, our method surpasses competitive baselines on IMAGENET that rely on the full 1.2M-image training set as a reference.

## 1 Introduction

Deep Neural Networks (DNNs) are typically developed under a closed-world assumption. When these models encounter unfamiliar inputs from the open world, they may face Out-of-Distribution (OOD) samples, which can severely disrupt system operations. In safety-critical applications such as autonomous driving Kitt et al. (2010) and healthcare Schlegl et al. (2017), identifying and handling these OOD inputs is crucial. For instance, a self-driving car may fail to detect objects on the road that are not included in the training set, which could lead to an accident.

To distinguish OOD samples from In-Distribution (InD) data, a rich line of OOD detection algorithms has recently been developed. Current OOD detection methodologies, as surveyed by Yang et al. (2021c), predominantly fall into categories that either heavily depend on the availability and characteristics of InD data or require exposure to OOD samples during training. OOD Exposure methods involve collecting external OOD samples during training (e.g., OE Hendrycks et al. (2018a), MCD Yu & Aizawa

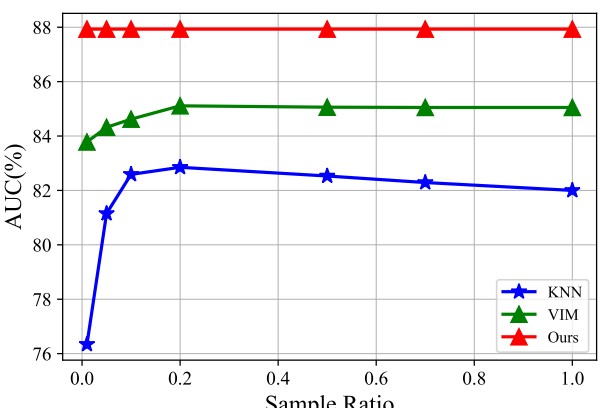

Figure 1: OOD detection performance with different sampling ratios on IMAGENET training set (1.2 million images). TTA-based method is InD-independent and thus not affected by the sampling ratio. With only 25 TTAs, TTA-based method outperforms KNN Sun et al. (2022) and VIM Wang et al. (2022), which rely on the entire training set.

(2019), UDG Yang et al. (2021b)) to help the detector learn the InD/OOD boundary. While simple, their effectiveness is limited as they cannot reliably detect OOD samples unseen during this exposure phase. InD-dependent methods, such as those using Mahalanobis distance Lee et al. (2018), K-Nearest Neighbors

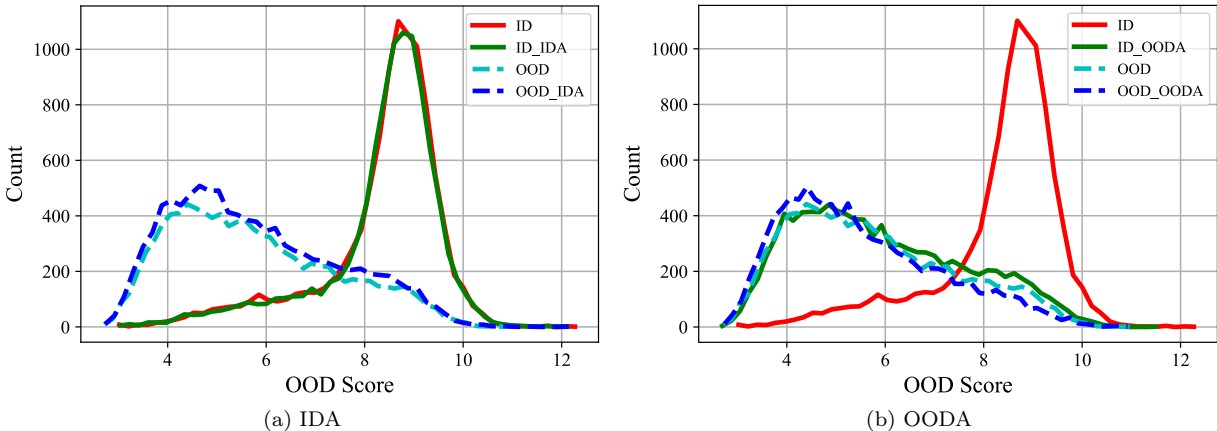

(a) IDA        (b) OODA

Figure 2: The influence of InD augmentation (IDA) and OOD augmentation (OODA) on the distribution of OOD Score.

(KNN) Sun et al. (2022), or Virtual Logit Matching (VIM) Wang et al. (2022), utilize known InD data as a reference set or for statistical estimation. However, a critical drawback is their performance sensitivity to the quantity and quality of this InD data, as illustrated in Fig. 1. This reliance on extensive, well-curated InD datasets significantly curtails their practical deployability and scalability. In contrast, InD-independent methods (e.g., MSP Hendrycks & Gimpel (2016), ML Hendrycks et al. (2019a), Energy Liu et al. (2020), ODIN Liang et al. (2017)) devise scoring functions based on model outputs. While more user-friendly, their standalone performance often requires further improvement to meet the demands of critical applications.

The use of data augmentation to enhance OOD detection, with methods like Mixup Zhang et al. (2017), CutMix Yun et al. (2019), and PixMix Hendrycks et al. (2022), has primarily focused on the training phase. While some work by He et al. (2022) has hinted at the utility of Test-Time Augmentations (TTAs) for OOD detection, a comprehensive investigation into TTA's impact and its potential as a primary mechanism for OOD detection remains notably scarce. This oversight represents a missed opportunity, given TTA's inherent potential for data-efficient, input-specific analysis.

**This paper studies test-time augmentation (TTA) as a self-referential signal for out-of-distribution (OOD) detection.** The key idea is to probe a model locally around a *single* test input using mild, controlled perturbations and to quantify the stability of the resulting representations/scores. This perspective enables strong OOD detection without storing any InD reference set and without using explicit OOD exposure during training.

Our main contributions are:

1. **IDA/OODA taxonomy for TTA in OOD detection.** We empirically show that mild, feature-preserving *In-Distribution Augmentations* (IDAs) consistently improve OOD detection, while aggressive *OOD Augmentations* (OODAs) often degrade it by perturbing InD samples too strongly (Fig. 2 and Tables 1, 2). We further provide practical guidance for selecting IDAs via perceptual similarity.

2. **Single-sample, reference-free detector via self-neighborhood probing.** We introduce a simple plug-and-play OOD detector that compares a test input to its own IDA views in an embedding space, using the $k$-th largest similarity as an "ID-ness" score. Unlike conventional KNN that searches over a stored InD dataset, our method performs the search within the test sample's TTA neighborhood.

3. **Sequential masking as a scalable IDA family.** We propose sequential masking to generate many informative, low-perceptual-change IDAs from one input, enabling strong detection with a small number of TTAs (Fig. 4).

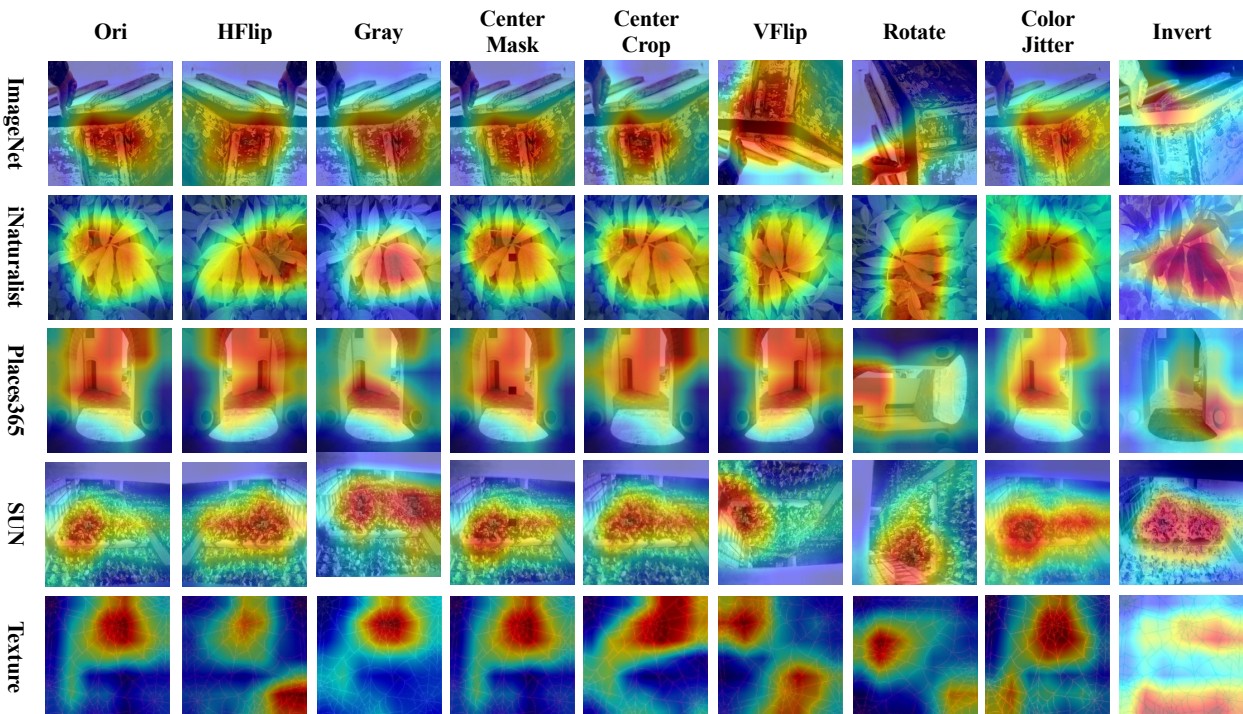

Figure 3: Heatmaps of IDAs and OODAs for InD and OOD. The visualization technology we use is the improved Grad-CAM, which uses a global average of the gradients backpropagated from the Energy score to compute the weights of the feature maps.

4. **Comprehensive evaluation.** Across CIFAR-10 and IMAGENET benchmarks, multiple architectures, robustness settings, and combinations with existing methods (e.g., ReAct/ASH), we demonstrate that the proposed TTA-based detector is competitive while removing the need for large InD reference sets (Fig. 1 and Section 5).

## 2 Background and Related Works

### 2.1 OOD Detection Strategies

**Data-Reliant Approaches (OOD Exposure and InD-Dependent):** Many methods attempt to improve OOD detection by either exposing the model to known OOD samples during training (e.g., Outlier Exposure (OE) Hendrycks et al. (2018a), or using synthetic/auxiliary OOD data Lee et al. (2017); Yu & Aizawa (2019); Wu et al. (2021); Yang et al. (2021b)) or by heavily relying on In-Distribution (InD) data. The latter category includes techniques that use InD data as a reference set for comparisons (e.g., Mahalanobis distance Lee et al. (2018), KNN Sun et al. (2022)), for statistical estimations (e.g., VIM Wang et al. (2022)), or for training auxiliary model components designed to aid OOD detection (e.g., ConfBranch DeVries & Taylor (2018), CSI Tack et al. (2020), MOS Huang & Li (2021), VOS Du et al. (2022)).

*Position Perspective on Data-Reliant Approaches:* The fundamental limitation of these strategies is their dependence on external data. OOD exposure methods are constrained by the availability and representativeness of chosen OOD samples, risking overfitting and failing to generalize to truly novel OOD inputs. InD-dependent methods are critically influenced by the quantity and quality of the InD dataset (as seen in Figure 1), rendering them less scalable and practical where large, curated InD sets are infeasible. Our advocacy for TTA stems from its potential to bypass these extensive data requirements.

**InD-Independent Approaches:** Seeking to reduce data dependency, InD-independent methods devise scoring functions based directly on a model's output for a given test sample, without reference to an

Table 1: OOD Detection Performance (AUROC) of TTAs on CIFAR-10. IDAs are markedly more effective than OODAs. Combining multiple IDAs (e.g., Hflip + Gray + CenterMask + CenterCrop) yields the best performance, motivating careful TTA selection.

| TTA Type | | OOD Datasets | | | | | | |
|---|---|---|---|---|---|---|---|---|
| | | Cifar100 | SVHN | Texture | Places365 | iSUN | LSUN | Average |
| IDA (Our Focus) | Hflip | 87.93 | 95.13 | 88.92 | 90.39 | 95.84 | 98.33 | _92.76_ |
| | Gray | 86.77 | 92.49 | 87.38 | 88.71 | 93.42 | 96.75 | 90.92 |
| | CenterMask | 87.43 | 95.07 | 87.89 | 88.49 | 94.24 | 97.99 | 91.85 |
| | CenterCrop | 87.17 | 95.27 | 89.10 | 90.24 | 95.77 | 98.06 | 92.60 |
| | Fourier Low Pass | 87.02 | 94.40 | 89.27 | 90.70 | 96.88 | 98.08 | 92.73 |
| | Hflip + Gray | 87.63 | 95.21 | 88.93 | 90.24 | 95.71 | 98.43 | 92.69 |
| | Hflip + Gray + CenterMask | 88.37 | 95.24 | 88.97 | 90.11 | 95.48 | 98.37 | _92.76_ |
| | Hflip + Gray + CenterMask + CenterCrop | 88.80 | 94.78 | 89.55 | 90.69 | 95.91 | 98.12 | **92.97** |
| OODA (Less Effective) | Vflip | 55.88 | 46.14 | 42.71 | 61.53 | 59.89 | 62.06 | 54.70 |
| | Rotate | 53.55 | 50.55 | 45.88 | 61.54 | 58.83 | 58.38 | 54.79 |
| | ColorJitter | 65.87 | 61.88 | 61.03 | 70.80 | 69.05 | 70.65 | 66.55 |
| | Invert | 73.94 | 77.58 | 68.42 | 77.15 | 77.33 | 83.02 | 76.24 |
| | Fourier High Pass | 59.24 | 53.38 | 48.91 | 71.84 | 63.96 | 64.16 | 60.25 |

InD dataset. This includes approaches utilizing softmax/logit scores (MSP Hendrycks & Gimpel (2016), ML Hendrycks et al. (2019a)), energy values Liu et al. (2020), or other model-internal signals like gradients or activation patterns (ODIN Liang et al. (2017), GRAM Sastry & Oore (2020), DICE Sun & Li (2022), GradNorm Huang et al. (2021), ReAct Sun et al. (2021), ASH Djurisic et al.).

## 2.2 Augmentation in OOD Detection: The Untapped Potential of TTA

Data augmentation is widely recognized for its benefits in regularizing models during the *training phase* to improve uncertainty estimation and generalization, which indirectly aids OOD detection. Techniques like Mixup Zhang et al. (2017), AugMix Hendrycks et al. (2019b), CutMix Yun et al. (2019), PixMix Hendrycks et al. (2022), and YOCO Han et al. (2022) manipulate training data, and some even search for optimal training augmentations via reinforcement learning Mohseni et al. (2021). A systematic study by Geiping et al. (2022) on the effect of training-phase data enhancement on OOD generalization found that aggressive augmentations yield diverse features but unstable gains, whereas mild augmentations lead to more consistent features and stable, albeit potentially smaller, gains. This distinction between mild and aggressive augmentations during training provides a valuable precursor to understanding TTA. We argue that the principles distinguishing mild versus aggressive augmentations, as noted by Geiping et al. (2022) for training, are even more critical at test time and form a cornerstone of effective TTA-based OOD detection, as we will elaborate in Section 3. The current lack of focus on TTA for OOD detection represents a significant missed opportunity for advancing the field towards more practical and efficient solutions.

# 3 A Closer Look at Test-Time Augmentation for OOD Detection

Data augmentation has long been recognized as a cornerstone technique for improving model generalization during training, enabling networks to learn invariances and robust representations. However, its role during the *testing phase* for OOD detection has remained relatively underexplored. We show that Test-Time Augmentation (TTA) provides an effective, self-referential signal for OOD detection by probing model behavior through localized variations of a test input, without requiring any global InD reference data. This section analyzes when and why TTA helps, and provides practical guidance for selecting effective test-time augmentations.

## 3.1 A Principled Taxonomy of Test-Time Augmentations

Inspired by Geiping et al. (2022), who classified training-time augmentations based on their impact on image expression, we argue that a similar classification is crucial for test-time data augmentation to harness its OOD detection capabilities. We categorize TTAs as follows:

Table 2: OOD Detection Performance of TTAs on IMAGENET. The detection performance of IDA is much higher than that of OODA, and using multiple augmentations leads to the optimal performance.

| TTA | | OOD Datasets | | | | |
|---|---|---|---|---|---|---|
| | | iNaturalist | Places365 | SUN | Texture | Average |
| IDA (Our Focus) | Hflip | 80.67 | 73.46 | 73.22 | 70.46 | 74.45 |
| | Gray | 85.89 | 69.18 | 75.03 | 66.67 | 74.19 |
| | CenterMask | 81.58 | 68.68 | 76.11 | 73.25 | 74.91 |
| | CenterCrop | 80.11 | 74.24 | 75.49 | 74.22 | 76.02 |
| | Fourier Low Pass | 82.36 | 71.58 | 73.46 | 71.92 | 74.83 |
| | Hflip + Gray | 82.75 | 74.44 | 74.93 | 71.22 | 75.83 |
| | Hflip + Gray + CenterMask | 82.30 | 73.37 | 76.47 | 72.41 | 76.13 |
| | Hflip + Gray + CenterMask + CenterCrop | 83.63 | 73.97 | 76.76 | 73.33 | **76.92** |
| | Hflip + CenterMask | 80.97 | 71.81 | 76.96 | 72.88 | 75.66 |
| | Hflip + CenterCrop | 83.13 | 74.46 | 75.23 | 72.53 | 76.34 |
| | Gray + CenterMask | 80.19 | 70.75 | 77.99 | 73.18 | 75.53 |
| | Gray + CenterCrop | 82.28 | 72.97 | 76.70 | 73.36 | 76.33 |
| | CenterMask + CenterCrop | 81.27 | 71.75 | 77.75 | 73.95 | 76.18 |
| | Gray + CenterMask + CenterCrop | 81.81 | 72.12 | 78.16 | 73.98 | 76.52 |
| OODA (Less Effective) | Vflip | 43.42 | 63.61 | 75.30 | 55.72 | 59.51 |
| | Rotate | 43.38 | 63.61 | 75.26 | 55.46 | 59.43 |
| | ColorJitter | 70.28 | 59.21 | 71.80 | 57.91 | 64.80 |
| | Invert | 76.55 | 58.89 | 82.01 | 62.40 | 69.96 |
| | Fourier High Pass | 84.37 | 66.37 | 72.08 | 56.41 | 69.81 |

Table 3: LPIPS of different data augmentations. This table supports our argument for selecting mild TTAs (IDAs), which generally yield lower LPIPS scores, indicating less perceptual change and aligning with their superior OOD detection performance shown in Table 1. Masking, a key component of our proposed TTA strategy, achieves the lowest LPIPS.

| LPIPS | Hflip | Gray | Mask | Crop | Vflip | Rotate90 | ColorJitter | Invert |
|---|---|---|---|---|---|---|---|---|
| CIFAR-10 | 0.048 | 0.1184 | **0.0052** | 0.0171 | 0.075 | 0.082 | 0.1618 | 0.2368 |
| IMAGENET | 0.2961 | 0.2466 | **0.01** | 0.1425 | 0.5839 | 0.6312 | 0.4484 | 0.5656 |

- **In-Distribution Augmentations (IDAs)**: These are mild TTAs that we posit are key to effective TTA-based OOD detection. They do not significantly alter core image features. Examples include horizontal flip (HFlip), grayscale conversion, small-size center masking, large-size center cropping, and Fourier low-pass filtering.

- **Out-of-Distribution Augmentations (OODAs)**: These are aggressive TTAs that drastically change image features, such as vertical flip (VFlip), large rotations, ColorJitter, Invert, and Fourier high-pass filtering. We argue these are generally detrimental or uninformative for TTA-based OOD detection.

### 3.2 The Differential Impact of IDA vs. OODA on OOD Score Distributions

A key observation is that IDAs and OODAs have fundamentally distinct effects on OOD score distributions. As illustrated in Fig. 2, IDAs have a negligible effect on the score distribution of InD data, while subtly modifying the distribution of OOD data. This differential behavior is precisely what TTA can exploit. In contrast, OODAs induce a significant distribution shift in InD data itself, causing it to resemble OOD data and thereby confounding the detection task.

### 3.3 Performance Implications: The Superiority of IDAs for TTA-based Detection

The practical consequences of these differing impacts directly support our advocacy for IDA-focused TTA strategies. Across both CIFAR-10 and IMAGENET benchmarks (Tables 1 and 2), IDAs consistently outperform OODAs by substantial margins. On CIFAR-10, individual IDAs achieve average AUROC scores exceeding 90%—horizontal flip reaches 92.76% and Fourier low pass achieves 92.73%—while the best OODA (Invert)

manages only 76.24%. Geometric OODAs like Rotate and VFlip collapse to near-random performance at 54.79% and 54.70% respectively. Similar patterns emerge on IMAGENET, where center crop achieves 76.02% AUROC while Rotate and VFlip drop to 59.43% and 59.51%, representing performance gaps exceeding 16 percentage points.

Furthermore, employing multiple complementary IDAs significantly enhances detection through synergistic effects. The combination of Hflip + Gray + CenterMask + CenterCrop achieves the highest performance on both datasets—92.97% on CIFAR-10 and 76.92% on IMAGENET —demonstrating consistent improvements over individual augmentations. Even two-way combinations like Hflip + CenterCrop (76.34% on IMAGENET) outperform single augmentations, suggesting that different IDAs capture complementary aspects of distribution shift. This multi-augmentation strategy leverages diverse invariance properties: geometric (Hflip), color (Gray), and spatial locality (CenterMask, CenterCrop), creating comprehensive probes of model behavior while OODAs destroy the discriminative signals necessary for reliable detection.

### 3.4  Identifying Effective IDAs: Perceptual Similarity as a Guiding Principle

To operationalize the selection of IDAs and provide practical guidelines for practitioners, we leverage the key insight from Geiping et al. (2022) that milder augmentations have less disruptive impact on learned image features. The Learned Perceptual Image Patch Similarity (LPIPS) metric Zhang et al. (2018) serves as a valuable quantitative tool for this assessment, measuring the perceptual distance between original and augmented images through deep feature representations. As demonstrated in Table 3, IDAs consistently yield lower LPIPS scores than OODAs across both CIFAR-10 and IMAGENET, signifying greater perceptual similarity to the original and thus less disruptive impact on semantic content.

Crucially, we observe a strong negative correlation between LPIPS scores and OOD detection performance (compare Tables 3 and 1). Augmentations with lower perceptual distances tend to exhibit superior detection capabilities. Masking, notably achieving the lowest LPIPS scores (0.0052 for CIFAR-10, 0.01 for IMAGENET), emerges as particularly effective for OOD detection. This correlation provides a practical heuristic for identifying promising IDAs without extensive empirical search. While grayscale transformation presents an interesting exception with relatively high LPIPS despite strong performance—potentially due to LPIPS being sensitive to color changes versus structural preservation—the overall trend supports favoring perceptually similar augmentations.

### 3.5  Visualizing IDA's Efficacy Through Activation Analysis

To further demonstrate the impact of IDA and OODA on image features, we show the heat maps of common IDAs and OODAs on large-scale datasets in Fig. 3. The visualization results of CIFAR-10 are not shown because its resolution is too low. It can be observed that OODA has a great influence on the features of both InD and OOD data. IDA will not change the high thermal area of InD, while the high thermal area of OOD will be affected by IDA. Based on the observation of a large number of visualization results, we have obtained the following empirical conclusions:

- OOD data has a larger proportion of high thermal regions than InD data, that is, the useful features of OOD are more dispersed.

- IDAs do not change the high thermal region of InD, but they will change the high thermal region of OOD. And OODAs have an impact on the features of both InD and OOD. Therefore, IDA can be used for OOD detection, and OODA cannot be used for OOD detection.

- No single IDA was able to cause changes in the high thermal regions of all OOD data. Horizontal flip is an effective TTA for OOD Detection, but the third row of Fig. 3 (Places365) shows that horizontal flip does not have as much impact on the heatmap as other TTAs.

These visualizations of activation heatmaps reveal a key phenomenon: the OOD heatmap is noticeably affected by IDAs, while the InD heatmap remains largely unaffected. This differential perturbation in salient

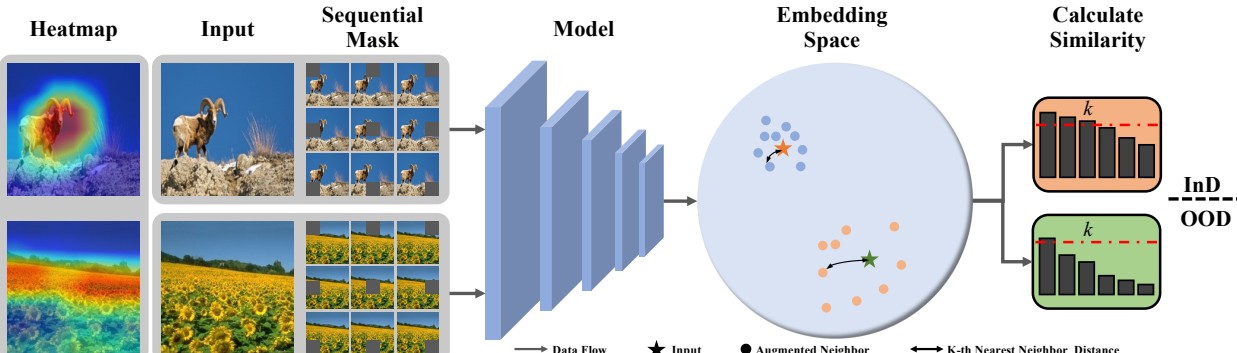

Figure 5: Overview of TTA-based method for OOD detection. We first perform a sequential mask for the input image. Next, the input image and corresponding TTAs are fed into the model to obtain embeddings. Then the $k-$th largest similarity between the input image and the TTAs embedding is selected as the ID score. If the score exceeds the threshold, it is detected as InD.

regions for OOD versus InD samples under mild augmentation explains why IDAs can be leveraged for OOD detection, as evidenced in Table 1.

***Takeaway:*** *IDAs—unlike OODAs—create distinguishable behavioral differences (e.g., in activation heatmaps and OOD score distributions) between InD and OOD samples while largely preserving InD characteristics. This makes IDA-based TTA strategies suitable for OOD detection and provides a data-efficient alternative to reference-set-based methods.*

## 4 TTA-based Method

This section presents a simple TTA-based OOD detection methodology that leverages a test input's *own* augmented views, enabling data-efficient detection without storing any InD reference set.

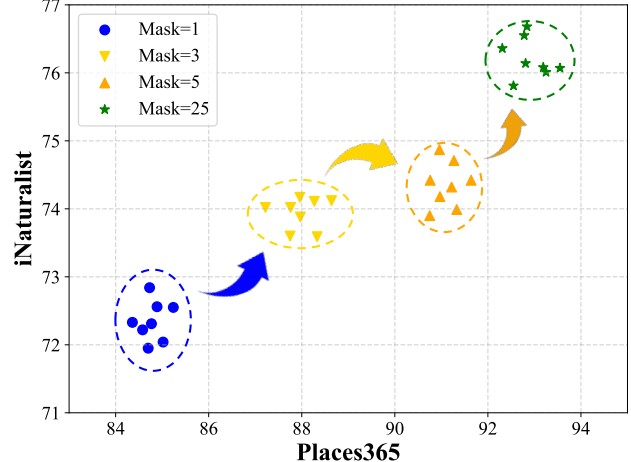

Figure 4: Detection performance with different numbers of masks on IMAGENET. $x$ and $y$ axes indicate the detection performance on Place365 and iNaturalist. The detection performance improves as the number of masks increases.

### 4.1 Preliminary

Let $\mathcal{X}$ denote the input space and $\mathcal{Y}$ denote the label space. Given a pre-trained classifier $F : \mathcal{X} \to \mathcal{Y}$ trained on InD data drawn from distribution $P_{in}$, the goal of OOD detection is to determine whether a test sample $x \in \mathcal{X}$ is drawn from $P_{in}$ or from an unknown distribution $P_{out}$. We denote by $\phi(x) \in \mathbb{R}^d$ the representation used for detection (e.g., logits, softmax scores, or their combination) extracted from $F$. Formally, we seek a scoring function $g : \mathcal{X} \to \mathbb{R}$ and a threshold $\tau$ such that:

$$h_\tau(x) = \begin{cases} \text{InD}, & \text{if } g(x) \geq \tau \\ \text{OOD}, & \text{if } g(x) < \tau \end{cases} \tag{1}$$

The challenge lies in designing $g$ without access to $P_{out}$ and, ideally, with minimal dependence on stored samples from $P_{in}$.

### 4.2 Design Objective

In contrast to previous methods that heavily rely on InD data Sun et al. (2022); Lee et al. (2018) or OOD exposure, the primary objective here is to *design an effective OOD detection method that leverages TTA*

---

**Algorithm 1** TTA-based OOD Detection

---

1: **Input:**
   Input image $x$, Representation extractor $\phi(\cdot)$, Number of masks $K$, Mask size $m \times m$, Similarity threshold $\tau$, KNN parameter $k$
2: **Output:** OOD detection result
3: Let $\mathcal{M}$ be an empty set                    # *Set of masked images*
4: Let $H$ and $W$ be the height and width of $x$
5: Let $s_h = \lfloor H/m \rfloor$ and $s_w = \lfloor W/m \rfloor$                    # *Stride*
6: **for** $i = 1$ **to** $K$ **do**
7:    Let $h = (i \bmod s_h) \times m$
8:    Let $w = \lfloor i/s_h \rfloor \times m$
9:    Let $x_i$ be $x$ with an $m \times m$ mask at position $(h, w)$
10:    $\mathcal{M} \leftarrow \mathcal{M} \cup \{x_i\}$
11: **end for**
12: Define a function **ComputeSimilarity**$(z_1, z_2)$:
13:    **return** $\frac{z_1 \cdot z_2}{\|z_1\|\|z_2\|}$                    # *Cosine similarity*
14: Let $\mathcal{Z}$ be an empty set                    # *TTA embeddings*
15: Let $z = \phi(x)$                    # *Original embedding*
16: **for all** $x_i \in \mathcal{M}$ **do**
17:    $\mathcal{Z} \leftarrow \mathcal{Z} \cup \{\phi(x_i)\}$
18: **end for**
19: Let $S$ be an empty set                    # *Similarities*
20: **for all** $z_i \in \mathcal{Z}$ **do**
21:    Let $s_i = $ **ComputeSimilarity**$(z, z_i)$
22:    $S \leftarrow S \cup \{s_i\}$
23: **end for**
24: Sort the elements in $S$ in descending order
25: Let $s_k$ be the $k$-th largest value in $S$                    # *KNN similarity*
26: **if** $s_k \geq \tau$ **then**
27:    **return** InD
28: **else**
29:    **return** OOD
30: **end if**

---

*to achieve true In-Distribution data independence and obviates the need for OOD data during training or reference.* The approach explores the relationship between a sample and its carefully chosen TTAs, exploiting this for OOD detection without altering the underlying classifier, thus ensuring model-agnostic, plug-and-play capability.

### 4.3  Core Idea: Local Neighborhood Probing via TTA.

The central tenet of this TTA-centric approach is to construct a robust scoring function for OOD detection by analyzing the relationship between an input sample and its own TTAs. This fundamentally differs from traditional KNN, which searches within the feature space of an entire, often massive, InD training dataset. Instead, this method focuses on searching within the local neighborhood of the input sample, as defined by its TTAs. This inherently leads to a data-efficient and InD-independent solution. The success of this hinges on selecting appropriate IDAs, as argued in Section 3.

### 4.4  TTA Strategy: Sequential Masking

Our arguments in Section 3 emphasize the need for effective IDAs. While conventional IDAs are limited, and simple combinations do not always yield improvements (e.g., Hflip + Gray vs. Hflip alone in Table 1), our investigation points to a highly effective novel TTA strategy: ***Sequential Mask***. This approach, leveraging

the observation that masking has a minimal perceptual impact (lowest LPIPS in Table 3), applies masks to images in a sequential manner. This generates a substantial number of similar IDAs from a single input. The efficacy of this strategy is evident in Fig. 4, which shows a clear trend: OOD detection performance on IMAGENET (across Place365 and iNaturalist datasets) progressively improves with an increasing number of IDAs generated via sequential masking (using mask sizes of 8x8 on CIFAR-10 and 44x44 on IMAGENET). This demonstrates that a well-designed TTA strategy can create a rich set of informative views from the input itself.

### 4.5 Framework Overview

Fig. 5 illustrates the TTA-based OOD detection framework. Given an input sample $x$, we first generate multiple IDA views $\{x_1, \ldots, x_K\}$ using sequential masking. We then extract representations $\phi(x)$ and $\{\phi(x_i)\}_{i=1}^K$ and compute cosine similarities between $\phi(x)$ and each $\phi(x_i)$. After ranking these similarities, we use the $k$-th largest similarity as an "ID-ness" score and compare it against a threshold $\tau$ (calibrated on InD validation data, independent of any OOD data):

$$g(x) = s_{(k)}(x), \qquad h_\tau(x) = \begin{cases} \text{InD}, & \text{if } g(x) \geq \tau \\ \text{OOD}, & \text{if } g(x) < \tau \end{cases} \tag{2}$$

Here, $s_{(k)}(x)$ denotes the $k$-th largest similarity between $\phi(x)$ and $\{\phi(x_i)\}_{i=1}^K$. The value of $k$ can be selected using standard validation techniques Hendrycks et al. (2018b).

### 4.6 Algorithm

Algorithm 1 presents the complete pipeline of our TTA-centric OOD detection method, which consists of three carefully designed components that work synergistically to achieve robust detection without requiring any InD reference data.

**Sequential Mask Generation (Lines 3-11):** Given an input image $x \in \mathbb{R}^{H \times W \times C}$ where $H$, $W$, and $C$ denote height, width, and channels respectively, we generate $K$ masked versions through the sequential masking strategy. Our approach divides the image into a grid of non-overlapping patches, each of size $m \times m$ pixels. Unlike random masking strategies that may lead to uneven coverage and inconsistent detection performance, our sequential masking ensures comprehensive and uniform exploration of the image space. Specifically, we iterate through the grid positions in a predefined order (e.g., raster scan), masking one patch at each step to create augmentation $x_i$. This systematic approach guarantees that: (1) each region of the image is evaluated for its contribution to the OOD score, (2) the masking pattern is deterministic and reproducible, and (3) the computational cost is predictable and bounded by $K = \lfloor H/m \rfloor \times \lfloor W/m \rfloor$.

The mask size $m$ is carefully selected based on the input resolution to balance between maintaining sufficient image context and creating meaningful perturbations. For lower resolution inputs like CIFAR-10 (32×32), we use smaller masks (8×8) to preserve image structure, while for higher resolution inputs like IMAGENET (224×224), larger masks (44×44) are employed to create more substantial augmentations that better reveal distribution shifts.

**Embedding Extraction and Similarity Computation (Lines 12-23):** For both the original image $x$ and its masked versions $\{x_1, x_2, ..., x_K\}$, we extract representations $\phi(\cdot)$. The choice of representation space is important for detection performance (see the ablation in Fig. 8). We compute cosine similarities between $\phi(x)$ and each $\phi(x_i)$:

$$s_i = \frac{\phi(x) \cdot \phi(x_i)}{||\phi(x)||_2 \cdot ||\phi(x_i)||_2}, \quad i = 1, 2, ..., K \tag{3}$$

The use of cosine similarity, rather than Euclidean distance, provides scale-invariance and better captures the angular relationships between embeddings, which proves more robust for OOD detection across diverse data distributions.

**KNN-based OOD Score Computation (Lines 24-30):** Our scoring mechanism employs a k-nearest neighbor approach within the set of similarity scores $\{s_1, s_2, ..., s_K\}$. After sorting these similarities in

Table 4: Comparison with competitive OOD detection methods on CIFAR-10. A is AUROC and F is FPR95, ↑ indicates larger values are better and vice versa. The **bolded** values are the best performance, and the _underlined italicized_ values are the second-best performance, the same below.

| AUC | Training Data | Cifar100 | | SVHN | | Texture | | Places365 | | iSUN | | LSUN | | Average | |
|---|---|---|---|---|---|---|---|---|---|---|---|---|---|---|---|
| | | **F↓** | **A↑** | **F↓** | **A↑** | **F↓** | **A↑** | **F↓** | **A↑** | **F↓** | **A↑** | **F↓** | **A↑** | **F↓** | **A↑** |
| MSP | | 56.29 | 88.11 | 40.67 | 94.36 | 48.74 | 91.13 | 51.96 | 89.24 | 37.80 | 94.03 | 28.59 | 95.91 | 44.01 | 92.13 |
| ML | | 49.65 | 88.09 | 28.08 | 95.32 | 41.33 | 91.06 | 42.52 | 89.87 | 26.36 | 95.21 | 13.55 | 97.58 | 33.58 | 92.86 |
| Energy | | 48.09 | 88.18 | 25.63 | _95.49_ | 39.77 | 91.15 | _40.59_ | 90.02 | 24.34 | 95.38 | 11.85 | 97.79 | 31.71 | 93.00 |
| ODIN | | 50.86 | 82.10 | _23.14_ | 92.69 | 38.44 | 87.07 | 42.99 | 85.58 | **15.33** | _95.82_ | _10.05_ | 97.56 | **30.13** | 90.14 |
| VIM | ✓ | 54.22 | 87.33 | **15.61** | 94.85 | **25.02** | **94.89** | 50.32 | 89.10 | 30.57 | 95.62 | 47.79 | 94.19 | 37.25 | 92.66 |
| KNN | ✓ | 51.90 | 90.27 | 35.32 | 95.31 | 40.30 | _93.86_ | 45.88 | _91.19_ | 28.86 | 95.70 | 28.23 | 96.00 | 38.42 | _93.72_ |
| GradNorm | | 73.54 | 60.13 | 65.14 | 68.62 | 73.24 | 57.31 | 68.55 | 66.90 | 62.01 | 70.00 | 41.70 | 82.57 | 64.03 | 67.59 |
| DICE | ✓ | 53.71 | 83.79 | 26.86 | 93.89 | 40.82 | 89.43 | 47.46 | 85.04 | 30.18 | 93.14 | **8.01** | **98.17** | 34.51 | 90.58 |
| GEN | | **45.21** | _88.60_ | 27.83 | 95.22 | 40.33 | 91.79 | **36.43** | 91.00 | 24.51 | 95.52 | 12.89 | 97.36 | _31.20_ | 93.25 |
| NAC | ✓ | _46.27_ | 88.37 | 32.92 | 94.32 | _35.47_ | 90.37 | 44.82 | 88.83 | 22.18 | 95.55 | 14.36 | 97.27 | 32.67 | 92.45 |
| ASH-P | | 53.02 | 85.52 | 38.73 | 94.44 | 38.26 | 89.46 | 45.64 | 87.15 | 24.64 | 94.36 | 18.55 | 97.35 | 36.47 | 91.38 |
| ASH-B | | 61.46 | 74.22 | 51.22 | 80.55 | 62.35 | 77.63 | 65.82 | 78.50 | 46.97 | 83.36 | 26.36 | 89.78 | 52.36 | 80.67 |
| ASH-S | | 50.15 | 84.27 | 27.64 | **96.17** | 39.77 | 89.55 | 46.78 | 84.72 | 25.69 | 94.17 | 16.18 | _98.13_ | 34.37 | 91.17 |
| Ours | | 47.36 | **90.79** | 30.39 | 95.17 | 37.07 | 93.50 | 43.23 | **91.29** | _17.46_ | **96.95** | 15.24 | 97.45 | 31.79 | **94.19** |

descending order to obtain $\{s_{(1)}, s_{(2)}, ..., s_{(K)}\}$ where $s_{(1)} \geq s_{(2)} \geq ... \geq s_{(K)}$, we select the $k$-th highest similarity as the OOD score:

$$\text{OOD-Score}(x) = s_{(k)} \tag{4}$$

This design choice is motivated by several considerations: (1) Using the $k$-th similarity rather than the maximum provides robustness against outlier augmentations that may accidentally preserve or destroy critical features, (2) It allows for controlled sensitivity adjustment through the parameter $k$, where smaller values yield more conservative detection, (3) Unlike mean or median aggregation, this approach maintains sensitivity to distribution tails, which is crucial for detecting subtle distribution shifts.

The final detection decision compares this score against a threshold $\tau$, which can be calibrated on InD validation data to satisfy application-specific false positive rate requirements.

### 4.7 Core Advantages of the TTA-Centric Framework

This TTA-based framework offers several practical advantages:

1. **True InD-Independence**: Crucially, this method does not require any prior knowledge or storage of InD data for reference. This is a stark contrast to traditional KNN Sun et al. (2022), Mahalanobis distance methods Lee et al. (2018), and VIM Wang et al. (2022), which are fundamentally dependent on InD data for reference sets or covariance estimation. Consequently, the performance of this TTA-based method remains unaffected by InD data characteristics (see Fig. 1), making it genuinely distributional assumption-free and highly data-efficient.

2. **OOD-Agnostic by Design**: The testing process relies solely on the input and its self-generated augmentations, without any dependence on prior knowledge or examples of OOD data.

3. **Model-Agnostic and Plug-and-Play**: The procedure only requires embeddings from a pre-trained classifier and does not modify the classifier's architecture or weights. This makes the method broadly applicable across diverse model architectures, from CNNs to Transformers like ViT Dosovitskiy et al. (2020), without model-specific reconfiguration, as it primarily relies on input masking.

## 5 Experiments

### 5.1 Experimental Setting

**ID Datasets.** Following the latest OOD benchmark Yang et al. (2022; 2021a), we chose CIFAR-10 Krizhevsky et al. (2009) and IMAGENET Krizhevsky et al. (2017) as the ID datasets. CIFAR-10 consists of 10 classes

Table 5: OOD Detection Performance on IMAGENET. The labeling is the same as Table 4.

| AUC | Training Data | NINCO | | SSB-hard | | iNaturalist | | OOD Datasets Places365 | | SUN | | Texture | | Average | |
|---|---|---|---|---|---|---|---|---|---|---|---|---|---|---|---|
| | | F↓ | A↑ | F↓ | A↑ | F↓ | A↑ | F↓ | A↑ | F↓ | A↑ | F↓ | A↑ | F↓ | A↑ |
| MSP | | 72.38 | 79.63 | 90.33 | 70.03 | 53.43 | 88.01 | 76.49 | 78.23 | 73.74 | 79.83 | 70.73 | 78.59 | 72.85 | 79.05 |
| ML | | 69.54 | 79.91 | 89.84 | 70.29 | 48.32 | 91.31 | 73.28 | 81.03 | 66.35 | 84.39 | 60.78 | 84.26 | 68.02 | 81.87 |
| Energy | | 70.22 | 79.14 | 93.56 | 69.86 | 50.54 | 90.96 | 74.01 | 80.60 | 65.02 | 84.52 | 58.69 | 84.57 | 68.67 | 81.64 |
| ODIN | | 76.58 | 76.87 | 91.54 | 71.00 | 42.12 | 90.95 | 70.38 | 81.28 | 61.89 | 84.40 | 50.74 | 85.52 | 65.54 | 81.67 |
| VIM | ✓ | 70.17 | 78.99 | 96.35 | 64.01 | 73.56 | 87.12 | 87.25 | 77.50 | 83.68 | 79.23 | 22.93 | **96.60** | 72.32 | 80.58 |
| KNN | ✓ | 75.83 | 77.90 | 98.86 | 57.71 | 63.89 | 85.60 | 88.84 | 71.65 | 75.46 | 77.90 | **14.27** | _96.47_ | 69.53 | 77.87 |
| GradNorm | | 73.30 | 72.55 | **83.30** | 67.71 | 24.30 | 94.13 | _68.10_ | 75.74 | 44.20 | 88.16 | 37.40 | 88.54 | 55.10 | 81.14 |
| DICE | ✓ | 79.00 | 76.46 | 88.30 | 66.57 | 32.10 | 93.06 | 76.10 | 79.54 | 51.20 | 86.24 | 43.10 | 88.01 | 61.63 | 81.65 |
| GEN | | 79.50 | 81.22 | 87.20 | 69.07 | 46.40 | 92.19 | 79.70 | 80.60 | 75.30 | 82.64 | 67.00 | 83.25 | 72.52 | 81.49 |
| NAC | ✓ | 73.62 | 78.47 | 86.52 | 68.26 | 36.31 | 93.52 | 70.33 | 78.53 | 53.24 | 88.81 | 49.75 | 88.14 | 61.63 | 82.62 |
| ASH-P | | 63.83 | 80.26 | 96.73 | 69.29 | 36.54 | 92.87 | 70.82 | 81.67 | 58.48 | 86.35 | 49.51 | 87.84 | 62.65 | 83.05 |
| ASH-B | | 60.32 | 81.95 | 86.17 | _71.23_ | _16.41_ | _97.40_ | 68.56 | **84.82** | 38.49 | **94.42** | 19.36 | 94.09 | **48.22** | **87.32** |
| ASH-S | | **58.65** | **82.77** | 95.74 | 68.11 | **14.87** | **98.06** | 64.32 | _83.09_ | 42.37 | _92.72_ | _16.08_ | 96.46 | _48.67_ | _86.87_ |
| Ours | | _59.33_ | _82.19_ | _85.81_ | **71.43** | 37.10 | 92.55 | 74.88 | 75.81 | _40.10_ | 91.82 | 35.37 | 91.54 | 55.43 | 84.22 |

of 32x32 color pictures, containing a total of 60,000 images, and each class contains 6000 images. Among them, 50000 images are used as the training set and 10000 images are used as the test set. IMAGENET is a large-scale dataset with 1000 classes, its training set contains 1.2 million images and its validation set contains 50,000 images. We resize all images to 224x224.

**OOD Datasets.** According to the existing OOD detection benchmarks Yang et al. (2022), we select six OOD datasets for both CIFAR-10 and IMAGENET. For CIFAR-10, the OOD datasets are Cifar100 Krizhevsky et al. (2009), SVHN Netzer et al. (2011), Texture Kylberg (2011), Places365 Zhou et al. (2017), iSUN Xu et al. (2015) and LSUN Yu et al. (2015), with Cifar100 being the near OOD and the rest being the far OOD. For IMAGENET, the OOD datasets used are NINCO Bitterwolf et al. (2023), SSB hard Vaze et al. (2022), iNaturalist Van Horn et al. (2018), Places365, SUN Xiao et al. (2010), Texture, where NINCO and SSB hard are near OOD and the rest are far OOD.

**Evaluation metrics.** We mainly use the following two metrics to evaluate OOD detection algorithms: 1) FPR95 measures the false positive rate (FPR) at which the true positive rate (TPR) is equal to 95%, a lower score indicates better performance. 2) AUROC measures the area under the receiver operating characteristic (ROC) curve, showing the relationship between TPR and FPR. The area under the ROC curve can be interpreted as the probability that a positive ID example has a higher detection score than a negative OOD example, with higher scores indicating better performance. In this paper, we use AUROC as the main metric.

**Backbones.** We use ResNet18 He et al. (2016) as the backbone for CIFAR-10. The model is trained for 200 epochs, with a batch size of 128. We use the cosine annealing learning rate Loshchilov & Hutter starting at 0.1. We train the models using stochastic gradient descent with momentum 0.9, and weight decay $5e - 4$. We use a ResNet50 He et al. (2016) backbone with resolution 224x224 for IMAGENET, and use the pre-trained weights from torchvision maintainers & contributors (2016) with a 76.13% accuracy.

**Baseline Methods.** We compare our methods with eleven baselines that do not require fine-tuning. They are MSP Hendrycks & Gimpel (2016), MaxLogit Hendrycks et al. (2019a), Energy Liu et al. (2020), ODIN Liang et al. (2017), VIM Wang et al. (2022), KNN Sun et al. (2022), GradNorm Huang et al. (2021), DICE Sun & Li (2022), GEN Liu et al. (2023) and ASH Djurisic et al., where ASH has three shaping algorithms (**P**runing, **B**inary and **S**cale). VIM and KNN require 50,000 and 200,000 InD data on CIFAR-10 and IMAGENET, respectively. See Appendix A.5 for baseline settings.

## 5.2 Demonstrating Superior Data Efficiency and Performance

The TTA-based method outlined in Section 4 (referred to as "Ours" in performance tables) targets strong OOD detection while removing the need for an InD reference set.

**On Cifar-10 Tasks:** Table 4 presents results on CIFAR-10 across six OOD datasets. Our sequential-masking-based detector achieves an average AUROC of 94.19%. This is notable compared to traditional KNN, which reaches 93.72% AUROC while requiring the full 50,000-image CIFAR-10 training set as a reference database.

Table 6: Robustness of OOD Detection Methods on CIFAR-10.

| Method | Adversarial Attacks | | | | OOD AUC |
| | FGSM | PGD | C&W | Average | |
|---|---|---|---|---|---|
| SimCLR (Ours) | 77.63 | 81.33 | 71.43 | **76.80** | 91.29 |
| Mask (Ours) | 66.10 | 83.34 | 58.54 | 69.33 | **94.19** |
| MSP | 86.37 | 22.26 | 79.33 | 62.65 | 92.13 |
| ML | 85.62 | 1.84 | 79.25 | 55.57 | 92.86 |
| Energy | 85.48 | 1.84 | 79.16 | 55.49 | 93.00 |
| ODIN | 88.01 | 6.56 | 79.40 | 57.99 | 90.14 |
| KNN | 22.82 | 71.52 | 50.63 | 48.32 | 93.72 |
| VIM | 58.56 | 83.65 | 63.50 | 68.57 | 92.59 |

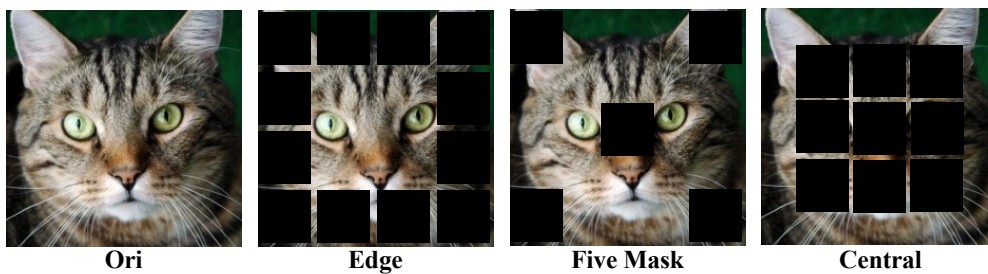

**Ori**      **Edge**      **Five Mask**      **Central**

Figure 6: The Visualization of Different Mask Strategies.

The gain is most visible on the near-OOD dataset Cifar100 (90.79% vs. 90.27% AUROC), suggesting that local self-neighborhood probing can capture subtle distribution shifts without stored references.

**On Large-Scale ImageNet Tasks:** The advantages become even more compelling when scaled to IMAGENET, as detailed in Table 5. Our method achieves an average AUROC of 84.22% across six challenging OOD datasets. In terms of data dependency, traditional KNN requires storing and searching through 1.2 million IMAGENET training images, while our approach generates only 25 masked augmentations per test sample. On particularly challenging near-OOD datasets (NINCO and SSB-hard), our method demonstrates competitive performance with 59.33% and 85.81% FPR95 respectively. While methods like ASH Djurisic et al. achieve higher average AUROC, they require dataset-specific configurations (ASH-P for CIFAR-10, ASH-B for IMAGENET). In contrast, our method uses the same scoring rule across datasets, with only resolution-dependent mask sizes.

**Direct Advantage Over ID-dependent Methods:** The fundamental difference between our approach and traditional ID-dependent methods manifests in several practical advantages. Methods like VIM Wang et al. (2022) require statistics estimated from (and often features stored for) the training set, while KNN Sun et al. (2022) necessitates maintaining a reference database. Our approach eliminates these requirements entirely by computing OOD scores solely from the input sample and its self-generated augmentations.

### 5.3 Versatility and Robustness of the TTA-Centric Paradigm

**Synergy with Existing Methods:** Table 8 demonstrates the remarkable synergy between our TTA approach and existing OOD detection techniques. When combined with ReAct Sun et al. (2021), our method's performance improves from 94.19% to 94.27% on CIFAR-10 and from 84.22% to 86.33% on IMAGENET. The combination with ASH-S yields even more impressive gains, achieving 94.86% on CIFAR-10 and 87.64% on IMAGENET. These results indicate that TTA-based approaches can serve both as standalone solutions and as powerful enhancement modules for existing methods. The consistent improvements across different base methods suggest that TTA captures complementary information about distribution shifts that traditional approaches may miss.

Table 7: Performance of Different TTA Strategies on CIFAR-10.

| Method | OOD Datasets | | | | | | |
| | Cifar100 | SVHN | Texture | Places365 | iSUN | LSUN | Average |
|---|---|---|---|---|---|---|---|
| Sequential Mask | 90.79 | 95.17 | 93.50 | 91.29 | 96.95 | 97.45 | **94.19** |
| Edge Mask | 90.90 | 95.00 | 93.19 | 91.43 | 97.04 | 97.25 | 94.14 |
| Central Mask | 88.00 | 95.41 | 89.72 | 88.78 | 96.04 | 98.15 | 92.68 |
| Five Mask | 90.26 | 95.77 | 92.30 | 91.31 | 96.63 | 98.00 | 94.05 |
| Ten Mask | 90.26 | 95.94 | 90.96 | 91.63 | 96.77 | 98.41 | 94.00 |
| FiveCrop | 86.35 | 93.22 | 90.00 | 90.11 | 95.84 | 97.03 | 92.09 |
| TenCrop | 86.84 | 94.12 | 90.51 | 90.50 | 96.14 | 97.42 | 92.59 |
| SimCLR | 86.00 | 92.37 | 88.78 | 90.01 | 94.16 | 96.43 | 91.29 |

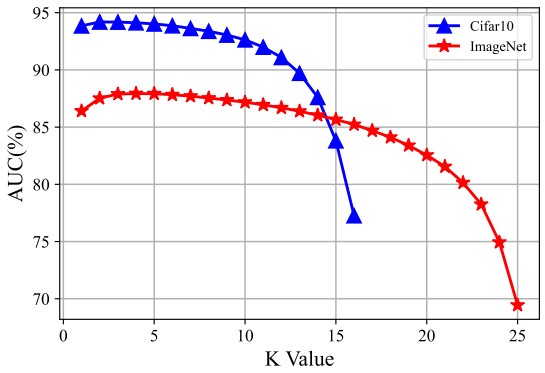

Table 8: Synergy between TTA and Other OOD Methods

| Method | Cifar-10 | ImageNet |
|---|---|---|
| Ours | 94.19 | 84.22 |
| ReAct | 92.66 | 83.96 |
| ASH-S | 91.17 | 86.87 |
| ReAct+Ours | 94.27 | 86.33 |
| ASH-S+Ours | **94.86** | **87.64** |

Figure 7: Detection performance of different k.

**Robustness to Adversarial Perturbations:** Table 6 presents robustness against three adversarial attack methods (FGSM Goodfellow et al. (2014), PGD Madry et al. (2017), C&W Carlini & Wagner (2017)). While logit/softmax-based methods (MSP, ML, Energy, ODIN) fail under PGD attacks with performance dropping below 25% AUROC, our sequential mask approach maintains 83.34% AUROC. When using diverse augmentations from SimCLR Chen et al. (2020), we obtain the highest average robustness (76.80%) across all attacks, with a trade-off in clean OOD detection performance (91.29% vs. 94.19%).

### 5.4 Ablation Studies

We conduct extensive ablation studies to validate design choices and characterize robustness across configurations. Note that the ablation studies on IMAGENET only include far OOD.

#### 5.4.1 Impact of TTA Strategy Selection

The choice of TTA strategy fundamentally determines the effectiveness of our approach. Fig. 6 illustrates five different masking strategies we evaluated. Table 7 presents their comparative performance on CIFAR-10. Sequential masking achieves the highest average AUROC of 94.19%, validating our systematic approach to generating augmentations. Edge masking, which preserves central image content, performs second-best at 94.14%, supporting our argument that mild augmentations (IDAs) are crucial for effective OOD detection. In contrast, central masking, which obscures primary image content, shows degraded performance at 92.68%, confirming that overly aggressive augmentations can harm detection capability. The comparison with traditional augmentation strategies (FiveCrop and TenCrop) further validates our masking approach. These crop-based methods achieve only 92.09% and 92.59% AUROC respectively, likely because cropping reduces the available image information more drastically than masking. SimCLR augmentations, while effective for contrastive learning, yield 91.29% AUROC, suggesting that augmentations optimized for representation learning may not directly transfer to OOD detection tasks.

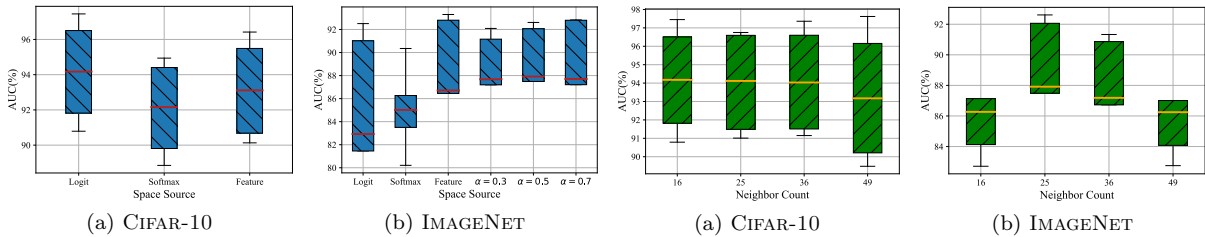

Figure 8: Detection performance using different spaces.

Figure 9: Detection performance using different mask sizes.

### 5.4.2 Analysis of k-Nearest Neighbor Selection

Fig. 7 examines the sensitivity to the $k$ parameter in our KNN-based scoring. Unlike traditional KNN which often requires large $k$ values and shows high sensitivity to this choice, our method performs optimally with small $k$ values ($k = 2$ for CIFAR-10, $k = 4$ for IMAGENET). The performance curves show a gradual decline as $k$ increases, rather than the sharp drops often seen in traditional KNN. This stability stems from our use of augmentations from the same input, which naturally have more consistent similarity patterns than samples from diverse training data.

The robustness to $k$ selection represents a significant practical advantage. While traditional KNN may require extensive validation to select optimal $k$ values for each dataset and can fail catastrophically with poor choices, our method maintains strong performance across a range of $k$ values, simplifying deployment and reducing the need for dataset-specific tuning.

### 5.4.3 Source Space Analysis

Fig. 8 investigates the impact of different embedding spaces on detection performance. For CIFAR-10, using logit space alone achieves optimal performance, while IMAGENET benefits from combining logit and softmax spaces with equal weighting (0.5 each). This difference likely stems from the increased complexity and number of classes in IMAGENET, where softmax probabilities provide additional discriminative information. Importantly, even suboptimal space choices maintain strong performance—using only softmax on CIFAR-10 still achieves over 92% average AUROC, demonstrating the robustness of our approach to this design choice.

### 5.4.4 Mask Size and Sample Count Optimization

Fig. 9 presents a detailed analysis of mask size impact on detection performance. For CIFAR-10 with 32x32 images, we evaluate mask sizes of {8, 6, 5, 4} pixels, generating {16, 25, 36, 49} masked samples respectively. The optimal configuration uses 8x8 masks with 16 samples, achieving 94.19% average AUROC. For IMAGENET's 224x224 images, mask sizes of {54, 44, 37, 32} pixels are tested, with 44x44 masks and 25 samples proving optimal at 84.22% AUROC.

The performance plots reveal important insights: (1) Too few augmentations provide insufficient evidence for reliable OOD detection, (2) Too many augmentations with smaller masks begin to significantly alter image semantics, degrading performance, (3) The method shows remarkable robustness—even with suboptimal configurations, CIFAR-10 maintains above 93% AUROC and IMAGENET exceeds 82% AUROC. This robustness to hyperparameter selection enhances the practical applicability of our approach.

### 5.4.5 Architectural Flexibility and Model-Agnosticism

Table 9 demonstrates our method's consistent performance across five diverse architectures on IMAGENET. The method achieves strong results on both CNN architectures (ResNet50: 87.92%, DenseNet121: 87.75%, WideResNet101: 87.21%) and Vision Transformers (ViT-b-16: 86.43%, Swin-t: 84.73%). This architectural agnosticism is particularly valuable for practical deployment, as organizations can apply our method to their existing models without architectural modifications or retraining.

Table 9: Performance of Our Method with Different Architectures on IMAGENET.

| Architectures | OOD Datasets | | | | |
|---|---|---|---|---|---|
| | iNaturalist | Places | SUN | Texture | **Average** |
| ResNet50He et al. (2016) | 92.61 | 75.78 | 91.88 | 91.39 | 87.92 |
| DenseNet121Huang et al. (2017) | 92.03 | 74.25 | 91.53 | 93.21 | 87.75 |
| WideResNet101Zagoruyko & Komodakis (2016) | 94.41 | 84.28 | 86.36 | 83.80 | 87.21 |
| Vit-b-16Dosovitskiy et al. (2020) | 92.70 | 82.81 | 84.71 | 85.43 | 86.43 |
| Swin-tLiu et al. (2021) | 90.79 | 81.54 | 82.17 | 84.40 | 84.73 |

Table 10: Detection Performance of Different OOD Detection Methods on Swin Transformer.

| AUC (%) | NINCO | SSB-hard | iNaturalist | Places365 | SUN | Texture | Avg |
|---|---|---|---|---|---|---|---|
| MSP | 80.22 | 71.14 | 89.94 | 77.93 | 79.65 | 80.57 | 79.91 |
| ML | 81.15 | 68.20 | 89.07 | 73.06 | 75.58 | 79.08 | 77.69 |
| ODIN | 62.65 | 63.14 | 70.57 | 46.30 | 55.13 | 65.47 | 60.54 |
| Energy | 77.14 | 68.47 | 84.99 | 67.47 | 70.88 | 76.44 | 74.23 |
| VIM | 81.03 | 69.08 | 91.34 | 76.44 | 77.52 | 87.54 | 80.49 |
| KNN | 79.44 | 64.17 | 87.59 | 77.18 | 76.49 | 88.28 | 78.86 |
| GradNorm | 45.52 | 49.98 | 38.70 | 26.41 | 32.78 | 35.46 | 38.14 |
| DICE | 41.20 | 57.20 | 32.60 | 32.53 | 35.55 | 70.80 | 44.98 |
| GEN | 80.66 | 68.04 | 90.68 | 80.50 | 81.64 | 82.32 | 80.64 |
| NAC | 76.58 | 67.29 | 91.48 | 75.53 | 80.87 | 83.14 | 79.15 |
| ASH-B | 82.26 | 70.13 | 94.32 | 85.14 | 88.10 | 89.75 | **84.95** |
| ASH-S | 80.24 | 68.24 | 92.61 | 81.64 | 85.56 | 87.65 | 82.66 |
| ASH-P | 82.35 | 67.73 | 93.19 | 83.42 | 87.48 | 89.05 | 83.87 |
| Ours | 81.37 | 67.71 | 90.79 | 81.54 | 82.17 | 84.04 | 81.27 |

The slightly lower performance on Swin-t prompted further investigation. Table 10 shows that this performance drop is not unique to our method—all baseline methods experience degraded performance on Swin-t, with some methods like GradNorm dropping to 38.14% average AUROC. Despite this general trend, our method maintains competitive performance at 81.27%, outperforming several baselines including KNN (78.86%) and demonstrating the robustness of our approach.

### 5.4.6 Computational Efficiency Analysis

Beyond detection performance, our method offers substantial computational advantages. Traditional KNN requires $O(n \cdot d)$ space complexity to store $n$ training samples with dimensionality $d$, and $O(n \cdot d)$ time complexity per query for exhaustive search. Our approach reduces this to $O(m \cdot d)$ for both space and time, where $m$ is the number of augmentations (typically 25 for IMAGENET versus 1.2 million training samples). This represents a thousand-fold reduction in computational requirements, enabling real-time OOD detection even on resource-constrained devices.

In summary, the experimental evaluation and ablation studies highlight that self-neighborhood probing with carefully chosen TTAs can deliver competitive OOD detection while substantially reducing dependence on stored InD reference data.

## 6 Conclusion

This paper demonstrates that carefully chosen test-time augmentations (TTAs), particularly mild IDAs, provide a strong self-referential signal for OOD detection. We show that this signal can be turned into a simple plug-and-play detector that achieves competitive performance without storing large InD reference sets or using OOD exposure. In particular, with only a small number of TTAs per input, the proposed sequential-masking-based detector can match or surpass reference-set-based baselines that rely on millions of InD images. Future work includes designing more effective (potentially learnable) IDA families and improving the accuracy–efficiency trade-off of multi-view test-time probing.

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

# A    Supplementary Experimental Results

This appendix provides additional experimental results. All experiments follow the same evaluation protocol described in the main paper unless otherwise specified.

## A.1    Generalization to CIFAR-100 as In-Distribution Dataset

We conducted additional experiments using CIFAR-100 as the in-distribution dataset and evaluated on four OOD test sets: CIFAR-10, SVHN, Texture, and Places365. As shown in Table 11, our method consistently outperforms all baselines across every OOD test set, achieving an average AUROC of 86.09% compared to the next best method ASH-P at 85.00%.

## A.2    Adversarial Robustness Against Detector-Targeted Attacks

To evaluate robustness against adaptive adversarial attacks that target the OOD detector's decision boundary, we conducted experiments on CIFAR-10 using FGSM, PGD, and C&W. Table 12 reports AUROC under attack. While all methods degrade (OOD detectors are not designed for adversarial robustness), our method maintains a consistent advantage over baselines.

Table 11: OOD detection performance (AUROC %) with CIFAR-100 as the in-distribution dataset. Our method consistently outperforms baselines across all OOD test sets.

| Method | CIFAR-10 | SVHN | Texture | Places365 | Average |
|--------|----------|------|---------|-----------|---------|
| KNN | 79.15 | 89.32 | 82.41 | 80.23 | 82.78 |
| ViM | 80.21 | 90.15 | 83.56 | 81.02 | 83.74 |
| NAC | 80.89 | 90.67 | 84.33 | 81.78 | 84.42 |
| ASH-P | 81.43 | 91.28 | 85.12 | 82.15 | 85.00 |
| **Ours** | **82.56** | **92.14** | **86.45** | **83.21** | **86.09** |

Table 12: Adversarial robustness evaluation (AUROC %) on CIFAR-10 with attacks specifically targeting OOD detectors. Our method maintains substantial advantages over baselines under all attack types.

| Method | Clean OOD | FGSM | PGD | C&W |
|--------|-----------|------|-----|-----|
| ASH-P | 91.38 | 45.23 | 32.56 | 38.92 |
| VIM | 92.66 | 52.18 | 38.34 | 42.15 |
| **Ours** | **94.19** | **68.45** | **58.72** | **61.38** |

### A.3 Comparison with Perturbation-Based Methods

We compare our TTA-based method with the perturbation-based approach proposed by Chen et al. Chen et al. (2025) using CIFAR-10 as the in-distribution dataset. As shown in Table 13, our method improves the average AUROC from 91.16% to 94.19%.

Table 13: Comparison with perturbation-based method (AUROC %) on CIFAR-10. Our method outperforms Chen et al. Chen et al. (2025) across all OOD test sets with an average improvement of 3.03%.

| Method | CIFAR-100 | SVHN | Texture | LSUN | iSUN | Places365 | Avg |
|--------|-----------|------|---------|------|------|-----------|-----|
| Chen et al. Chen et al. (2025) | 87.24 | 93.45 | 90.32 | 94.23 | 93.18 | 88.56 | 91.16 |
| **Ours** | **90.79** | **95.17** | **93.50** | **97.45** | **96.95** | **91.29** | **94.19** |

### A.4 Comparison of TTA-Based OOD Detection Methods

Table 14 summarizes key differences between our method and existing TTA-based OOD detection approaches. A practical distinction is that prior methods typically require batch-level operations (e.g., accumulating multiple test samples or updating batch normalization statistics), while our method operates on individual samples.

### A.5 Comparison with RTL on CIFAR-10

We compare with the Real-Time Linear (RTL) method on CIFAR-10 in Table 15. Our method outperforms RTL across all OOD test sets, achieving an average AUROC of 94.19% compared to 92.17% for RTL.

## Experiments Details

**Software and Hardware.** All methods are implemented in PyTorch 1.13. We run all the experiments on NVIDIA GeForce RTX-3090 GPU.

**Details of augmentations.** In Table 1, the mask size of CenterMask is 4x4, and the size of CenterCrop cropped image is 30x30 and then resize to 32x32. Both Fourier high-pass filtering and low-pass filtering

Table 14: Comparison of TTA-based OOD detection methods. Our method uniquely operates on single samples, enabling real-time and streaming applications.

| Method | Requirement | Key Difference |
|---|---|---|
| RTL Fan et al. (2024) | Batch of test samples | Requires multiple inputs for linear regression |
| Zhang et al. Zhang et al. (2025) | Batch statistics | Needs batch normalization updates |
| **Ours** | Single sample | Operates on individual inputs |

Table 15: Comparison with RTL (AUROC %) on CIFAR-10. Our method outperforms RTL across all OOD test sets while offering single-sample operation capability.

| Method | CIFAR-100 | SVHN | Texture | LSUN | iSUN | Places365 | Avg |
|---|---|---|---|---|---|---|---|
| RTL Fan et al. (2024) | 88.52 | 93.78 | 92.18 | 94.76 | 94.27 | 89.52 | 92.17 |
| **Ours** | **90.79** | **95.17** | **93.50** | **97.45** | **96.95** | **91.29** | **94.19** |

preserve 90% of the high-pass or low-pass signals. The angle of rotation is 90. In ColorJitter, brightness is 0.4, contrast is 0.4, saturation is 0.4, hue is 0.1.

**Hyperparameters for Baselines.** For VIM, when feature spaces of dimension $N > 1500$, we set the dimension of the main space to $D = 1000$, otherwise set $D = 512$. For KNN, the dimension of the penultimate feature where we perform the nearest neighbor search is 512 and 2048 on CIFAR-10 and IMAGENET respectively, and we choose $k = 50$ following Yang et al. (2022) for detection.

**Hyperparameters for Adversarial Attacks.** We compare the robustness of our method to adversarial attacks with existing OOD detection methods in Table 5. The attack methods we use are FGSM, PGD, and C&W. Among them, the perturbation budget of FGSM is 0.05 ($\epsilon = 0.05$), and that of PGD and C&W is $8/255$ ($\epsilon = 8/255$). The number of PGD attack steps is 50, and the step size is 0.002. The maximum number of iterations for C&W is 1000.

## Hyper-parameters in Augmentations

Table 16 investigates the detection performance of rotation and ColorJitter with large disturbance degree. It can be seen that the detection performance of these two augmentations is poor i.e., both are OODA. Moreover, we investigate the detection performance for different rotation angles in the Table 18. Results show that when the angle is small, the detection performance is higher than when the angle is slightly larger. This matches the intuition. When the rotation angle is small, it does not change the image features and can be regarded as IDA; when the rotation angle reaches a certain number of degrees such that it changes the image features, it becomes OODA.

## Augmentation Used in the Training Phase

To test the effect of adding augmentation during the training phase on the detection of IDA and OODA, we design three sets of experiments in Table 17 to compare the detection performance of using horizontal flip and vertical flip as TTA on models trained with horizontal flip, vertical flip and no augmentation. It can be observed that the detection performance of horizontal flip is much better than that of vertical flip on the model trained without augmentation. In addition, the performance of vertical flipping is improved on the model trained with vertical flipping. However, it is still weaker than the performance of horizontal flip.

Therefore, since our approach is to compare the output similarity of samples and augmentations, adding some kind of augmentation during the training phase will make this augmentation more like IDA, but it will still not perform as well as a deterministic IDA (e.g., horizontal flipping). Furthermore, since we are using multiple augmentations with K-nearest neighbor search, adding some OODAs will only slightly decrease the overall performance.

Table 16: Detection performance of OODA under different parameters. For ColorJitter, the 4 numbers represent brightness, contrast, saturation and hue.

| InD Dataset | Rotate | | | ColorJitter | | | |
|---|---|---|---|---|---|---|---|
| | 90 | 180 | 270 | 0.1,0.1
0.1,0.1 | 0.2,0.2
0.2,0.1 | 0.3,0.3
0.3,0.1 | 0.4,0.4
0.4,0.1 |
| Cifar10 | 54.79 | 54.83 | 54.64 | 68.81 | 68.32 | 67.55 | 66.55 |
| ImageNet | 59.43 | 63.28 | 63.49 | 64.79 | 64.80 | 64.81 | 64.80 |

Table 17: Augmentation used in the training phase

| Training Augmentation | Hflip | Vflip |
|---|---|---|
| Hflip | 92.76 | 54.70 |
| No Aug | 90.09 | 56.18 |
| Vflip | 89.58 | 78.23 |

Table 18: Detection Performance of Rotation with different degrees.

| InD Dataset | Rotate Degree | | |
|---|---|---|---|
| | 5 | 15 | 30 |
| Cifar10 | 92.05 | 85.26 | 46.03 |
| ImageNet | 75.85 | 68.40 | 54.09 |

## Visualization

We visualize the samples with their masked augmentations in Fig. 10 (a), and it can be seen that there may be some kind of "non-ideal" mask that causes the InD and OOD and their enhancements to be far apart. However, the use of multiple IDAs makes the distance between the InD and its nearest neighbour significantly smaller than that of the OOD.

We also show the distribution of embedding similarity between images from different datasets and their 16 IDAs in Fig. 10 (b). It also shows that multiple IDAs will lead to a significant difference in the distribution of embedding similarity between InD and OOD.

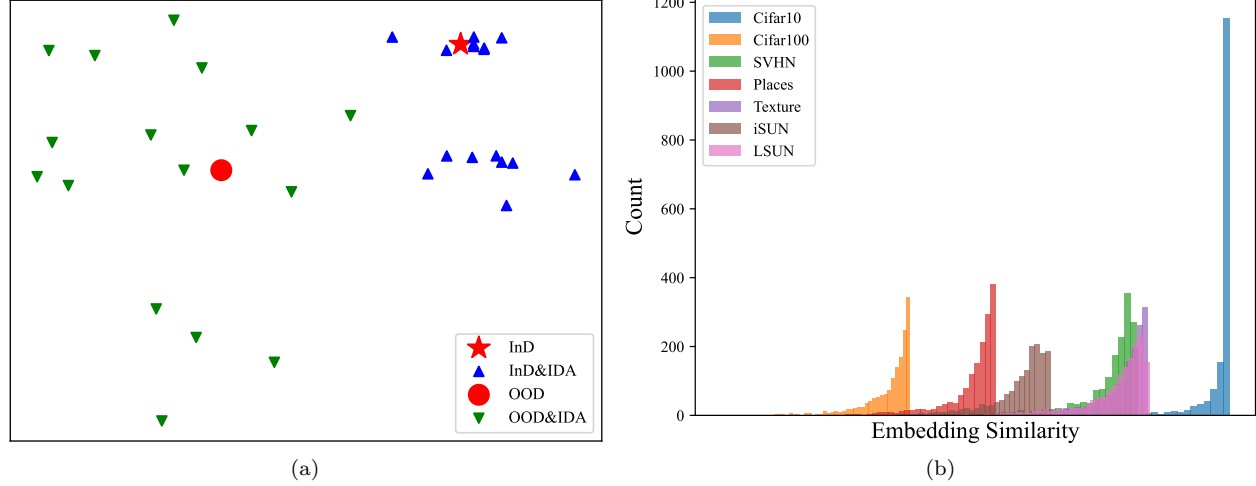

Figure 10: (a) Visualization of embeddings of InD and OOD, as well as their IDAs. It can be observed that the distance between InD and its nearest neighbor is much smaller than OOD. (b) The distribution of embedding similarity between images and their 16 IDAs. It shows that InD (Cifar10) has a higher embedding similarity to its IDAs than OOD.

