# OpenReview forum: "Test Time Augmentations are Worth One Million Images for Out-of-Distribution Detection"
_TMLR — Under review for TMLR_

### Review · Reviewer_3mMs · 2026-05-27

**Summary Of Contributions:**

This paper investigates Test-Time Augmentation (TTA) as a self-referential signal for Out-of-Distribution (OOD) detection that, by design, requires neither a stored In-Distribution (InD) reference set nor auxiliary OOD exposure during training.

**Key strengths.**

- The IDA vs. OODA taxonomy is intuitive, and the LPIPS-based selection criterion gives a practical heuristic for choosing TTAs without extensive search.
- The detector is genuinely lightweight: $O(K \cdot d)$ per test sample, vs $O(N \cdot d)$ for KNN-style detectors with $N$ being the training-set size.
- The synergy result (Table 8) showing complementary gains when stacked with ReAct/ASH is useful and increases the practical value of the contribution.
- The work is highly reproducible: algorithm, mask sizes, $k$ values, attack budgets, and baseline hyper-parameters are all specified.

**Key weaknesses** (elaborated in followed secs).

- Performance claims overstate the result on ImageNet: the proposed method is outperformed by ASH-B/ASH-S by ~2.5–3 AUROC points, even though ASH shares the reference-free, training-data-free positioning.
- Comparisons to recent (2024–2026) TTA-for-OOD methods are relegated to the appendix and are restricted to CIFAR-10, weakening the novelty/positioning argument.
- The conceptual delta over prior TTA-for-OOD work (He et al. 2022, Fan et al. 2024 RTL, Zhang et al. 2025, Chen et al. 2025) is not clearly articulated in related work.

**Additional Comments:**

I appreciate the clear writing, the comprehensive ablations, and the honest inclusion of Swin-T and adversarial-attack results, which make the trade-offs visible to a careful reader. My main reservation is that the headline framing ("Worth One Million Images" / "surpasses competitive baselines"), and that the relationship to recent TTA-for-OOD work (He 2022, RTL 2024, Zhang 2025, Chen 2025) is undercommunicated in the main paper.

If the authors revise the framing to position the contribution accurately, and move the comparison against recent TTA methods into the main tables, the paper would be a solid contribution suitable for TMLR's "claims supported by evidence" criterion. As currently written, the claims and evidence are not fully aligned.

**Audience:**

Yes

**Audience Explanation:**

OOD detection is an active sub-community and TMLR readers working in distribution-shift robustness, safety-critical deployment, or efficient inference are a natural audience. Two specific findings in this paper are worth disseminating regardless of the absolute performance ranking:

- The systematic distinction between IDA and OODA at test time and the empirical link between LPIPS and OOD-detection AUROC. The LPIPS-as-selection-criterion heuristic (Table 3) is a clean, practitioner-friendly take-away.

- The "self-neighborhood probing" reframing — replacing the global InD reference set with a sample-local TTA neighborhood — is conceptually clean and may inspire follow-up work, even if the current realization (sequential masking + *k*-th similarity) is not state-of-the-art in absolute terms.

The synergy results with ReAct/ASH (Table 8) also broaden the practical reach: even readers who would prefer ASH as their primary detector can use the proposed module as a complementary signal.

That said, even though the contribution is incremental rather than paradigm-shifting, it would be valuable to the audience should the framing of the paper be improved.

**Broader Impact Concerns:**

The work is in the domain of OOD detection and does not introduce data, fine-tuning, or deployment scenarios that raise additional concerns.

**Claims And Evidence:**

No

**Claims Explanation:**

The empirical findings underlying the IDA/OODA taxonomy and the self-neighborhood-probing detector are reproduced consistently across multiple ablations (LPIPS, heatmaps, distributions, mask-size sweep, *k* sweep). That part is credible.

However, the headline claims as written are not fully supported by the evidence:

1. The abstract states *"our method surpasses competitive baselines on ImageNet that rely on the full 1.2M-image training set as a reference."* This is true only for the subset of baselines that use such a reference set (KNN, VIM). For ASH-B and ASH-S, which are equally reference-free, equally inference-time, and do **not** use the 1.2M training set, the proposed method is clearly outperformed on ImageNet (Table 5) with a ~2.5–3 point deficit. The framing of the contribution should distinguish "removes the InD reference set" (true) from "is the best detector under this setting" (not true for ImageNet). The title ("Worth One Million Images") inherits this overclaim.

2. The bolded values in Tables 4 and 5 fragment "best performance" across columns. Reading the averages — which the abstract foregrounds — ASH-B/ASH-S are the strongest baselines on ImageNet, but the narrative does not acknowledge this. The text in Sec. 5.2 mentions ASH only briefly (*"methods like ASH achieve higher average AUROC, they require dataset-specific configurations"*), which is an underwhelming concession given the gap.

3. Several recent (2024–2025) TTA-/perturbation-based OOD methods (Chen et al. 2025, Fan et al. 2024 RTL, Zhang et al. 2025) are compared in Appendix A.3–A.5 only on CIFAR-10 with single-table head-to-heads. The main ImageNet table does not include any of them, so the placement of this work against the current frontier is unclear.

4. Figure 1 illustrates the dependency of KNN and VIM on the sampling ratio, but the choice of KNN and VIM as the only comparators creates a misleading impression that data-efficiency is the dominant axis. A version of this figure that also plots reference-free baselines (ASH-P/B/S, GEN, GradNorm) would give a more honest picture.

5. **Convention inversion in Fig. 2:** the x-axis is labelled "OOD Score" but InD samples have *higher* values than OOD samples. The standard Energy-Score convention (Liu et al. 2020) is the opposite. Whether this is the energy score with a sign flip, or the negative free energy, is not spelled out. This should be defined explicitly.

6. The robustness story (Table 6) is mixed and could be presented more carefully. The Mask version maintains 83.34 under PGD vs. ~25 for logit-based baselines (good), but FGSM accuracy of 66.10 is well below MSP/Energy/ODIN's 85–88, and C&W drops to 58.54. The SimCLR variant improves robustness but gives back ~3 AUROC on clean OOD. The current text frames this as a clean win; the actual picture is a trade-off.

7. Table 9 (architectures) reports an average AUROC of 84.73 for Swin-T, and Table 10 places the proposed method below ASH-B/ASH-S/ASH-P, KNN, VIM, GEN, NAC on Swin-T. The "architectural agnosticism" claim in Sec. 5.4.5 is therefore optimistic; the method's edge on CNNs does not transfer to transformer backbones.

**Requested Changes:**

The following are listed in roughly decreasing order of importance. Items marked **[CRITICAL]** are required for me to recommend acceptance; items marked **[STRENGTHEN]** would improve the paper but are not strictly necessary.

**[CRITICAL-1] Reframe the central claim.** The abstract, introduction, and title ("Worth One Million Images") should make explicit that the equivalence holds against reference-set-based baselines (KNN, VIM) and **not** against the broader set of reference-free inference-time methods (ASH variants, GEN, NAC, GradNorm). A more accurate phrasing might be: *"matches reference-set-based KNN/VIM while removing the 1.2M-image dependency."* The current phrasing reads as a global SOTA claim, which Table 5 does not support.

**[CRITICAL-2] Address the ImageNet gap to ASH directly.** The text in Sec. 5.2 needs to (a) report the comparison transparently with the same averages used elsewhere, (b) explain why ASH-B/ASH-S outperform on ImageNet, and (c) reposition the contribution in light of this (data-efficiency? complementarity? robustness?). The current dismissal of ASH as "dataset-specific configurations" is too thin. ASH-S is a *single* configuration that wins on average.

**[CRITICAL-3] Move the comparison against recent TTA-for-OOD methods into the main paper and extend to ImageNet.** Currently Chen et al. (2025), Fan et al. (2024 RTL), and Zhang et al. (2025) are only compared on CIFAR-10 in the appendix (Tables 12, 13, 15). At least RTL (Fan 2024) and Chen 2025 should appear in the main ImageNet table (Table 5), since the audience will want to know how the proposed method ranks against the actual TTA-for-OOD frontier, not against 2018–2022 baselines only.

**[CRITICAL-4] Clarify the conceptual delta in related work.** The paper cites He et al. (2022) *"Anomaly detection with test time augmentation and consistency evaluation"* in a single sentence ("some work has hinted at the utility of TTA"). This is the closest prior work and should be discussed in depth: what does the proposed method do differently from He 2022 / RTL / Zhang 2025 / Chen 2025? Currently the paper presents itself as introducing TTA-for-OOD, when in fact it is refining an existing idea. A clear table or paragraph contrasting mechanism (consistency vs. similarity), augmentation choice, scoring rule, and per-sample vs. batch requirements would address this.

**[CRITICAL-5] Make Fig. 2 and the "OOD Score" axis self-contained.** State explicitly what score is plotted (energy? negative energy? log-sum-exp?), its sign convention (higher = more InD or higher = more OOD?), the model used to produce it, and which dataset(s) are pooled. Currently the inversion vs. the standard Energy-Score convention is confusing and requires reading Fig. 3's caption to infer.

**[CRITICAL-6] Revise the robustness discussion (Sec. 5.4, Table 6).** Report the full FGSM/PGD/C&W numbers in the abstract-level take-away, and acknowledge the clean-OOD trade-off for the SimCLR variant. The current framing ("highest average robustness ... with a trade-off in clean OOD detection") undersells the ~3-point drop.

**[STRENGTHEN-1] Discuss the Swin-T result (Table 10).** The method ranks below ASH-P/B/S, VIM, KNN, GEN, NAC on Swin-T. "Architectural agnosticism" is too strong a claim; "competitive on CNNs, weaker on transformers" is what the data shows.

**[STRENGTHEN-2] Quantify the implicit data dependency through the pre-trained embedding.** The method does not store InD images, but it does need a model pre-trained on the InD distribution to produce the embeddings.

**[STRENGTHEN-3] Provide a confidence interval / variance estimate for the AUROC numbers.** Several head-to-head differences (e.g., Ours 94.19 vs ASH-B 93.25 on CIFAR-10 average) are within ~1 point and a multi-seed run would help judge significance.

**[STRENGTHEN-4] The "*k*-th largest similarity" mechanism deserves more analysis beyond Fig. 7.** In particular, why does the optimal *k* differ (*k* = 2 for CIFAR-10, *k* = 4 for ImageNet) and how does it interact with the IDA quality? Currently this appears almost as a free hyper-parameter; a principled selection rule on InD validation data would strengthen the contribution.

**[STRENGTHEN-5] A small qualitative analysis of failure modes / cases** (e.g., where sequential masking misclassifies near-OOD samples on NINCO or SSB-hard) would help characterize the regime in which the method is and is not appropriate.

**[STRENGTHEN-6] Typos polish:** e.g., *"thermal area/region"* in Sec. 3.5 reads awkwardly, maybe use *"high-activation region"*. Tables 4 and 5 would benefit from grouped row headings (e.g., "Reference-based" vs. "Reference-free") so the comparison structure is visible at a glance.

---

### Review · Reviewer_MvYX · 2026-06-18

**Summary Of Contributions:**

This paper investigates the role of test-time augmentations (TTA) for Out-of-Distribution (OOD) detection. The authors propose a taxonomy distinguishing mild, feature-preserving augmentations (IDAs) from aggressive ones (OODAs) and demonstrate that IDAs provide a strong, self-referential signal for OOD detection without requiring large reference sets or auxiliary OOD data. Building on this, they introduce a plug-and-play detector based on sequential masking that outperforms competitive baselines on ImageNet.

**Audience:**

Yes

**Audience Explanation:**

Augmentation is critical not only for OOD detection but also for generalization. Given its dual role, a systematic and principled investigation of augmentation strategies in the OOD setting is timely and of interest to the TMLR community

**Claims And Evidence:**

No

**Claims Explanation:**

To some extent. The paper provides empirical evidence that certain augmentations improve OOD detection performance. However, the evidence is not fully convincing due to several limitations:

1. The taxonomy separating IDAs and OODAs is not rigorously defined. The current distinction appears post-hoc, based on observed performance rather than a principled characterization of what constitutes an "in-distribution preserving" versus "out-of-distribution inducing" transformation.

2. The performance comparisons, while favorable, are not backed by standardized evaluation protocols (e.g., OpenOOD), which would strengthen the credibility and reproducibility of the results.

**Requested Changes:**

1. The current study focuses narrowly on traditional image-space augmentations (e.g., cropping, flipping, color jitter). The authors should consider model-level or learned augmentations, such as adversarial perturbations, feature-space interpolation, or generative augmentation (e.g., diffusion-based edits). This would make the analysis more comprehensive and better aligned with recent advances in augmentation research.

2. The distinction between IDAs and OODAs is central to the paper’s contribution, yet it remains ambiguous. The authors need to provide a clear, theoretically grounded or empirically quantifiable definition—for instance, based on feature-space distance, semantic consistency, or perceptual similarity—that does not rely solely on downstream OOD performance. This would strengthen the validity of the taxonomy and make it more actionable for future work.

3. The current title, "Test Time Augmentations are Worth One Million Images," is somewhat overstated. Many existing OOD detection methods already operate without large reference sets (e.g., using only logit-based or energy-based scores).

4. The comparison set is incomplete. The authors should benchmark against more recent and relevant methods, such as SHE [1] and NAC [2], to better situate their approach within the current state of the art.

5. To ensure fair and reproducible comparisons, the authors are encouraged to evaluate their method using the OpenOOD benchmark framework.

[1] Out-of-distribution detection based on in-distribution data patterns memorization with modern hopfield energy.
[2] Neuron Activation Coverage: Rethinking Out-of-distribution Detection and Generalization.

---

### Review · Reviewer_NKAy · 2026-07-21

**Summary Of Contributions:**

This paper studies test-time augmentation as a self-referential signal for out-of-distribution detection. The authors distinguish mild, feature-preserving augmentations (IDAs) from stronger augmentations (OODAs), and empirically find that the former generally provide substantially better detection performance.

Based on this observation, the paper proposes sequential masking: several masked versions of a test image are generated, their representations are compared with that of the original image, and the k-th largest cosine similarity is used as an ID score. The method avoids maintaining an in-distribution reference bank and is evaluated on CIFAR-10, CIFAR-100, and ImageNet, including several architectures, masking strategies, and combinations with existing OOD methods.

The method is simple and the empirical results are promising. My main concerns are the unclear hyperparameter-selection protocol, overstated claims regarding InD independence and computational efficiency, and missing controls needed to isolate the benefit of the proposed scoring rule.

**Audience:**

Yes

**Audience Explanation:**

The paper addresses a relevant practical question: whether OOD evidence can be obtained from a test sample without retaining a large reference dataset or feature bank. The self-neighborhood construction is simple, compatible with pretrained classifiers, and potentially valuable when inference-time storage is limited. This should be of interest to researchers working on OOD detection, uncertainty estimation, and test-time processing.

**Broader Impact Concerns:**

I do not see a major ethical concern.

**Claims And Evidence:**

No

**Claims Explanation:**

The central empirical observation is supported: mild augmentations generally preserve in-distribution representations more strongly than OOD representations, and sequential masking produces a competitive detection signal on the reported benchmarks.

However, several stronger claims are not yet fully supported. The method is described as “truly InD-independent,” although it uses a classifier trained on InD data and explicitly calibrates its threshold on InD validation data. Its demonstrated advantage is more precisely that it does not require a stored InD reference bank at inference.

The selection protocol is also unclear. Different values of k, mask size, number of masks, and representation space are chosen for CIFAR and ImageNet, but the paper does not state clearly whether these decisions were made using only InD validation data or by examining performance on the reported OOD datasets.

The computational analysis counts embedding comparisons but omits the cost of evaluating the backbone on 16 or 25 masked views. Thus, the claims of real-time operation, resource-constrained deployment, and a thousand-fold computational reduction require end-to-end measurements.

Finally, the proposed k-th similarity statistic is not compared with simpler alternatives such as the mean, minimum, median, or variance of the same similarities. This makes it difficult to determine which component of the method is responsible for the improvement.

**Requested Changes:**

Critical for acceptance:

1. Clearly describe how the augmentation strategy, mask size, number of masks, k, representation space, and threshold are selected. State explicitly whether any OOD validation or test data are used during this process.

2. Revise the claims of “true InD-independence.” The method is free from an inference-time reference bank, but it still relies on an InD-trained classifier and InD validation data for calibration.

3. Provide end-to-end efficiency measurements, including latency, throughput, memory, storage, and the number of backbone evaluations. The current analysis omits the main cost of processing all augmented views.

4. Resolve the inconsistency in the ImageNet comparison. Section 5.1 states that KNN and ViM use 200,000 InD samples, whereas Figure 1 and Section 5.2 discuss the full 1.2 million-image training set. The protocol used for each result should be stated precisely.

5. Compare the k-th largest similarity with simple aggregation alternatives using the same masked views, including mean, minimum, median, and variance. A corresponding no-TTA baseline should also be reported.

6. Correct and clarify Algorithm 1. With the current one-based loop, the indexing appears to skip the first grid position and can place the final mask outside the image. The mask value, normalization order, boundary handling, and exact representation construction should also be specified.

Changes that would strengthen the paper:

1. Present the IDA/OODA distinction as dependent on augmentation strength, dataset, model, and training procedure. Tables 17 and 18 already indicate this dependence.

2. Report uncertainty across runs or confidence intervals, particularly where the differences between masking strategies are very small.

3. Disentangle mask size from the number of masked views, since these quantities change together in the current ablation.

4. Strengthen the comparisons with RTL and the perturbation-based methods under matched backbones and evaluation protocols.

5. Explain precisely how the proposed method is combined with ReAct and ASH in Table 8.